



**Effect of the silica content of diatom prey on the production, decomposition and**
**sinking of fecal pellets of the copepod *Calanus sinicus***
Hongbin Liu*, Chih-Jung Wu
Division of Life Science, The Hong Kong University of Science and Technology,
Clear Water Bay, Kowloon, Hong Kong
* Corresponding author
Email: liuhb@ust.hk
FAX: (852)23581552



**Abstract**

13        The effects of changing the amount of silica in the cell wall of diatom prey, on

the production, decomposition rate and sinking velocity of fecal pellets of the
calanoid copepod, *Calanus sinicus*, were examined. Using different light intensities to
control the growth of the diatom *Thalassiosira weissflogii* also led to the
accumulation of different amounts of biogenic silica. Copepods were then fed with
either low (~1600 cells $L^{-1}$) or high (~8000 cells $L^{-1}$) concentrations of this diatom.
Copepods fed on a high concentration of diatoms with high silica content, exhibited a
lower grazing rate and lower fecal pellet production rate than those fed on a high
concentration of diatoms with low silica content. However, there was no difference in
either the grazing or fecal pellet production rates at low prey concentrations with high
or low silica content. The size of the fecal pellets produced was only affected by the
prey concentration, and not by the silica content of prey. In addition, the degradation
rate of the fecal pellets was much higher for copepods fed a low-silica diet than for
those fed on a high-silica diet. Significantly lower densities and sinking rates only
occurred in the fecal pellets of copepods fed a low-silica diet and a low prey
concentration. Calculating the L-ratio (the ratio of degradation rate:sinking rate) for
each group indicated that the fecal pellets produced by copepods fed on highly
silicified diatoms are likely to transport both biogenic silica and organic carbon to the
deep layer; whereas those produced following the consumption of low-silica diatoms
are likely to decompose in the mixing layer.






**Introduction**


In the marine environment, zooplankton fecal pellets constitute the main vehicle
for transporting biogenic elements to the sediments, although a substantial proportion
of this flux is recycled or repackaged in the water column by microbial decomposition
and zooplankton coprophagy (Turner, 2002; 2015). Diatoms are the most abundant
phytoplankton, and they represent the main component in the diet of zooplankton in
marine environments. Studies show that zooplankton with a diatom diet usually
produce fecal pellets that sink faster than those on other diets (Feinberg and Dam,
1998). Dagg et al. (2003) reported that the contribution of fecal pellets to the flux of
particulate organic carbon (POC) and biogenic silica (bSi) is higher during the spring
diatom bloom than during the summer within the Antarctic Polar Front region.
Similarly, Goldthwait and Steinberg (2008) reported an increase in mesozooplankton
biomass and fecal production and flux inside cyclonic and mode-water eddies.
However, González et al. (2007) reported a negative correlation between the vertical
carbon flux of diatoms and the production of fecal material in a time-series study in
the upwelling waters off Chile.
The quantity and characteristics of the fecal pellets produced by zooplankton
depend on several factors. The pellet production rate is reported to be affected by the
rate of ingestion and assimilation efficiency (Butler and Dam, 1994; Besiktepe and
Dam, 2002). It has also been demonstrated that the type of diet can affect the
characteristics of the fecal pellets produced; including size, density and sinking rates
(e.g., Feinberg and Dam, 1998 and ref. therein). In addition, the decomposition rate of
pellets varies with water temperature, as well as with both microbial and metazoan
activity (Poulsen and Iversen, 2008; Svensen et al., 2012). Factors that contribute to
the sinking velocity of the pellets include size, density and shape, all of which can
vary dramatically both among different zooplankton species and within the same



zooplankton species feeding on different types of prey (Fowler and Small, 1972;
Turner, 1977; Feinberg and Dam, 1998). Turbulence in the water column, the
presence or absence of a peritrophic membrane, and the production of microbial gas
within a peritrophic membrane might also affect the sinking rate of pellets (Honjo and
Roman, 1978; Bathmann et al., 1987). Indeed, the sinking rate and decomposition rate
are the two most important parameters used, to determine whether a pellet will or will
not be successfully transported into deeper water before its contents are degraded. For
example, a slowly sinking pellet is more likely to decompose and become part of the
recycled materials before it exits the euphotic zone (Dagg and Walser, 1986).

The cell wall (frustrule) of diatoms is composed of two silicate shells, which are

believed to act as a defense mechanism to prevent ingestion by grazers (Pondaven et
al., 2007); thus different levels of silicification of the frustrule might affect the grazing
rate of copepods (Friedrichs et al., 2013, Liu et al., in revision). The silica content of
the cell wall of diatoms is not only species-specific, but it is also affected by
environmental parameters such as light, temperature, salinity, pH, nutrients and trace
metals (Martin-Jézéquel et al., 2000 and ref. therein; Claquin et al., 2002; Vrieling et
al., 2007; Herve et al., 2012; Liu et al., in revision). Although the frustule has no
nutritional value for zooplankton, it is thought to provide ballast, which is especially
advantageous when the fecal pellets are sinking. Hence, pellets with a high diatom
biomass generally exhibit higher levels of export of POC (Armstrong et al., 2002;
François et al., 2002; Klaas and Archer, 2002). Thus, the content of the zooplankton
diet (and therefore the type and concentration of ballast minerals ingested) might
strongly affect the sinking velocity of the fecal pellets produced, and hence the
vertical flux of biogenic silica and carbon.

Most of the studies describing the production rates and characteristics of

copepod fecal pellets have focused on aspects such as food types (Feinberg and Dam,





1998), or the different periods of phytoplankton blooms (Butler and Dam, 1994).
There are currently no reports that describe the effect of the silica content of diatoms
on the production, degradation and sinking of fecal pellets. Liu et al. (under review)
recently demonstrated that the diatom *Thalassiosira weissflogii*, when grown at
different light levels, contains varying amounts of silica, and that the small calanoid
copepod *Parvocalanus crassirostris*, when fed on diatoms containing high levels of
silica exhibited a reduced feeding rate, and stagnant growth as well as low egg
production and hatching success. In this study we used the same diatom species with
different silica content as prey to study the characteristics of the fecal pellets produced
by the herbivorous copepod, *Calanus sinicus*.

**Materials and Methods**
**Copepod and prey culture conditions.** The herbivorous copepod *Calanus*
*sinicus* was collected from the coastal waters around Hong Kong in February 2013.
They were maintained on a 14 h light:10 h dark cycle at 23.5°C in 2 L glass
containers with 0.2 μm-filtered seawater. The copepods were fed a mixed algal diet
consisting of *Rhodomonas* sp. and *Thalassiosira weissflogii* at a concentration of
~5000 cells L$^{-1}$; this food suspension was supplied to the cultures twice a week and
the whole culture seawater was replaced every week. The copepods were maintained
for more than one month prior to the start of the experiment to ensure that all the
adults were grown in approximately the same conditions and were of approximately
the same age.
The diatom, *T. weissflogii*, was maintained in exponential growth in f/2 medium
(Guillard, 1975), under light intensities of either 15 μmol photons s$^{-1}$ m$^{-2}$ or 200 μmol
photons s$^{-1}$ m$^{-2}$ to generate cells with different cellular silica contents (Liu et al., under
review). The diatom cultures were transferred every 4 or 8 days for the high and low



light batches, respectively. After two transfers the amount of biogenic silica in the
diatom cells was measured using a modified version of the method described by
Paasche (1980), following the procedures described more recently by Grasshoff et al.

(1999).


**Experimental design.** Active adult female *Calanus sinicus* with intact
appendages were selected and starved for 24 hours before an experiment. A total of
seven experiments was conducted to determine fecal pellet production, degradation
and sinking, and in each experiment these parameters were measured both at low and
high food concentrations, and at high and low levels of silica contained in the diatom
prey (Table 1). In each experiment, the copepods were fed with the same species of
diatom (i.e., *T. weissflogii*), at either ca. 1600 cells $L^{-1}$ (low concentration), or ca.
8000 cell $L^{-1}$ (high concentration), the latter being above the food saturation level
according to Frost (1972). The abundance and volume of diatoms were measured
(triplicate subsamples) using a Beckman Coulter Z2 Particle Counter and Size
Analyzer.
In the fecal pellet production experiments, five replicate bottles containing one
copepod per bottle, and two control bottles without a grazer, were used. All the bottles
were filled with 100 ml freshly-prepared media consisting of 0.2 µm prefiltered
seawater and suspensions of the respective prey for each treatment. All incubations
were conducted at 23.5°C and in the dark for 24 hours. At the end of the incubation
period, a 2 ml sample was collected from each bottle and fixed with acid Lugol's at a
final concentration of 2%, for subsequent diatom quantification. The remaining water
was collected in a 50 ml polypropylene tube and fixed with glutaraldehyde at a final
concentration of 1%, for further quantification of the fecal pellets.
In order to obtain fresh pellets for the degradation experiments, two plastic



beakers were prepared for the high and low silica content prey. Each beaker contained
7-8 copepods and 700 ml culture medium, prepared as described for the production
experiments. After 12 hours of incubation (except for experiment # 3, which was
incubated for 18 hours), the medium was sieved through a 40 μm mesh to collect the
fecal pellets and then rinsed with autoclaved 0.22 μm filtered seawater. At least 20
intact fecal pellets were selected using a glass Pasteur pipette under a
stereomicroscope and poured into a 250 ml polycarbonate bottle containing 200 ml of
2 μm pre-filtered sea water taken from the field. The number of replicate bottles and
the incubation period of each experiment are show in Table 2. All the bottles were put
on a roller at 0.4 rpm in the dark at 23.5°C and then at the end of the respective
incubation times, the whole water of each bottle was collected in a plastic bottle and
fixed with glutaraldehyde at a final concentration of 1% for further fecal pellet
analysis.

Experiments to estimate the fecal pellet sinking rate were conducted by obtaining

fecal pellets using the degradation experiment procedure (described above) but with
an incubation time of 24 hours. After collecting all the fecal pellets from the beakers,
50 intact pellets were selected and suspended in 260 ml 0.2 μm prefiltered autoclaved
seawater. The fecal pellet sinking rate was measured using a SETCOL chamber (49
cm height, 2.6 cm inner diameter) made by 4 mm Plexiglas (Bienfang, 1981), filled
with well-mixed pellet-containing seawater. The chamber was allowed to settle for 6
min, and then the whole column of water was collected from outflow tubes in a
top-to-bottom order. The water was collected in a plastic bottle and fixed with
glutaraldehyde as described above, for subsequent fecal pellet analysis.

**Determining the number and size of fecal pellets.** The water samples
containing the fecal pellets in the 50-ml polypropylene tubes were allowed to settle



for 24 hours. The upper water was then removed smoothly and the remainder was
poured into the well of a 6-well plate and the number of pellets was counted using an
inverted microscope (Olympus IX51) at 100× magnification. Only intact fecal pellets
and fragments with end points were counted. The total number of fecal pellets was
then calculated to include all of the intact fecal pellets plus half of the pellet fragments.
Images of at least 30 intact fecal pellets were acquired with a CCD camera, after
which the length and width of each fecal pellet was measured and the volume was
calculated making the assumption that they are cylindrical in shape.

**Calculating the fecal pellet degradation rate.** The rate of degradation of the
fecal pellets was calculated from the loss of fecal pellet equation, described by:
$$N_t = N_0 e^{-rt}$$
where $N$ is the total number of fecal pellets in the incubation bottle at the
beginning ($N_0$) and end of the experiment ($N_t$); $t$ is the incubation time (in days); and $r$
is the degradation rate ($d^{-1}$). The degradation rate estimated in this study only
considered the effect of microbial organisms and assumed that the loss rate was
exponential.

**Calculating the fecal pellet sinking velocity.** The rate that fecal pellets sank
was calculated from the formula reported by Bienfang et al. (1982), which was
originally used to measure the average sinking rate of phytoplankton. Thus:
$$S = \frac{N_S}{N_T} \times \frac{L}{t}$$
where $S$ is the average sinking velocity; $L$ is the height of the sinking column; $t$ is the
duration of the trial; $N_T$ is the total number of fecal pellets within the settling water



volume; and Ns is the total number of fecal pellets that settled during the trial time.

In addition, the density of the fecal pellets was calculated using the

semi-empirical equation deduced by Komar (1980), as follows:
$$w_s = 0.079\frac{1}{\mu}(\rho_s - \rho)gL^2\left(\frac{L}{D}\right)^{-1.664}$$
where $w_s$ is the sinking velocity of the fecal pellets; $\mu$ and $\rho$ are the fluid viscosity and
density, respectively; $L$ and $D$ are the length and diameter of the fecal pellets,
respectively, assuming they are in the cylindrical shape; $g$ is the acceleration of
gravity; and $\rho_s$ is the density of fecal pellet.



## Results

### Grazing response



The cellular silica content of first and second generation *T. weissflogii* when
cultured at high and low light intensities is shown in Fig. 1. After two transfers the
cellular biogenic silica content was significantly different (t-test, $p<0.05$; Fig. 1) when
comparing the high light and low light culture conditions. The silica content of high
and low silica diatoms used in all the experiments was consistent and the differences
between the two treatments were all statistically significant (Table 1).
The grazing response of *C. sinicus* to diatoms with different silica contents
showed similar patterns between high (ca. 8000 cells ml$^{-1}$) and low (ca. 1600 cells
ml$^{-1}$) prey concentration (Fig. 2). At high concentrations of prey, *C. sinicus* grazed the
diatoms with low cellular silica content two times faster than when they had a high
silica content (t-test, $p<0.05$). The same trend was also observed at low concentrations
of the prey, although in this case the difference was not statistically significant. In
addition, the rate of clearance was significantly higher for the low silica prey than for
the high silica prey at both low and high prey concentrations (t-test, $p<0.05$). These
results indicate that the silica content of diatoms can affect the grazing activity of
copepods.

### Fecal pellet production


The rate of fecal pellet production varied both with the silica content and the
concentration of the prey (Fig. 3A). At a high prey concentration, *C. sinicus* that were
fed on low silica prey produced significantly higher amounts of fecal pellets ($192\pm32$
FP ind$^{-1}$ d$^{-1}$) than those fed on high silica prey ($113\pm47$ FP ind$^{-1}$ d$^{-1}$, $p<0.05$); which
corresponds well with the rate of ingestion (Fig. 2A and 3A). At a low prey
concentration, however, the production of fecal pellets by *C. sinicus* fed with the low



and high silica prey was not significantly different (Fig. 3A). In addition, the size of
the fecal pellets was only affected by the concentration of the prey, and not by the
silica content of the prey (Fig. 3B). Thus, the fecal pellets produced in the high
concentration of prey groups had a mean length and width of 582.4±98.7 μm and
72.5±4.5 μm, respectively, which are significantly larger than the size of those
produced in the low concentration of prey groups, which had an average length and
width of 352.4±54.7 μm and 59.6±6.8 μm, respectively (t-test, p<0.05).

**Fecal pellet degradation rate and sinking rate**

The degradation rate of fecal pellets was significantly different when the

copepods fed on diatoms with different silica content (Table 2). The degradation rate
of the fecal pellets produced from the low silica prey was approximately 4-5-fold
higher than that of the pellets generated from the high silica prey irrespective of the
prey concentration or the period of degradation incubation. In addition, the
degradation rate of the fecal pellets from low prey concentration was significantly
higher than ones from high prey concentration after an incubation period of 24 hr (p
<0.05, t-test). Furthermore, the degradation rate obtained following 48 h incubation
was significantly higher than that following just 24 h incubation (only high prey
concentration experiments) for both the high (p <0.05) and low (p< 0.01) silica prey
(Table 2), indicating an acceleration of degradation in the second day of incubation.

The sinking rate of fecal pellets was also different for the high and low prey

concentrations (Fig. 4). At a high concentration of prey, the sinking rates of the pellets
produced by the high and low silica prey (i.e., 3.05 and 3.13 cm min$^{-1}$, respectively),
were not significantly different. However, at a low prey concentration, the sinking
rate of pellets from the high silica content prey (i.e., 2.59 cm min$^{-1}$) was significantly
greater (t-test, p<0.01) than that of pellets from the low silica content prey (i.e., 0.53



cm min$^{-1}$). The average density of the fecal pellets was calculated as being
1.093-1.095 g cm$^{-3}$ at the high prey concentration, and 1.035-1.097 g cm$^{-3}$ at the low
prey concentration. The variation in the calculated density of fecal pellets is consistent
with the pattern of sinking rate, with the lowest density occurred in fecal pellets from
low silica prey at the low prey concentration (Fig. 4).

**Discussion**
The grazing activity of copepods varies not only with the concentration of the
prey but also with the nutritional quality of the prey. In our study, the grazing and
clearance rates determined with the varying food concentration, followed a similar
trend to that described in the literature (e.g., Frost, 1972). In addition, the grazing
activity was affected by the cellular silica content of the prey, as has been observed
with other copepod species (Liu et al., under review). Silicification is one of the
strategies that is used by diatoms to protect them from ingestion by grazers (Pondaven
et al., 2007). Friedrichs et al. (2013) examined the mechanical strength of the frustules
of three diatom species and measured the feeding efficiency of copepods on these
diatoms. Their results showed that the diatom species with the more weakly silicified
frustules and the highest growth rate was the least stable and was fed upon the most,
whereas the species with the most complex frustule exhibited the greatest stability and
was fed upon the least. Within the same species of diatom, different growth rates have
resulted in different amounts of silica in the frustule (Claquin et al., 2002). This
results in higher copepod ingestion and clearance rates for diatoms with a low silica
content when compared with those for diatoms with a higher silica content (Liu et al.,
under review). The results obtained in this new study are consistent with those
reported by Friedrichs et al. (2013) and Liu et al. (under review).
Previous studies indicate that while there is a linear relationship between the



total number of fecal pellets produced in unit time and the ingestion rate (Ayukai and
Nishizawa, 1986; Ayukai, 1990), there is a high level of variation among different
diets (Båamstedt et al., 1999 and the ref. therein; Besiktepe and Dam, 2002). In
addition, the size of the fecal pellets increases as the concentration of the food
increases, such that they reach a maximum size when the concentration of food is
above the saturation level (Dagg and Walser, 1986; Butler and Dam, 1994). Our
results confirmed these previous findings and demonstrated that the size of the pellets
produced was only affected by the concentration of prey, and they did not show any
significant differences when comparing prey of high and low cellular silica content.
Butler and Dam (1994) reported that when sufficient food was available, the size of
the fecal pellets varied with the nutritional quality (e.g., the C:N ratio) of the prey.
Since diatoms with different silica content (generated by varying the light intensity)
do not differ in their cellular C:N ratio (Claquin et al., 2002; Liu et al., under review),
they do not affect the size of the pellets produced.
The degradation rate and sinking velocity of the fecal pellets are highly
dependent on the characteristics of the pellets, which are in turn affected by the
quality and quantity of the food ingested (Feinberg and Dam, 1998; Turner, 2002;
2015 and ref therein). For example, it is known that the decomposition rate of the
fecal pellets is affected by diet, pellet size and the producer of the pellets (e.g., Shek
and Liu, 2010), but no research mentions the degradation rate of fecal pellets
produced by prey under different stoichiometric conditions. Hansen et al. (1996)
estimated the degradation rate of fecal pellets produced from diets of *Thalassiosira*
*weissflogii*, a diatom; *Rhodomonas baltica*, a nanoflagellate; or *Heterocapsa triquetra*,
a dinoflagellate. They showed that the fecal pellets produced from a diet of the diatom
species presented the slowest rate of degradation when compared with those produced
from diets of the nanoflagellate or dinoflagellate species. Similarly, Olesen et al.



(2005) compared the degradation rate of fecal pellets produced on a diet of the diatom,
*Skeletonema costatum*, or the nanoflagellate, *Rhodomonas salina*, and reported a
similar trend but higher degradation rates than Hansen et al. (1996). The relationship
between the surface:volume ratio and the degradation rate of fecal pellets was used to
explain the variation in the degradation rate of pellets produced with different diets.
Our results (Table 2) were higher than those reported by Hansen et al. (1996), which
were 0.024 d$^{-1}$ for *T. weissflogii*, but they showed a similar trend to those summarized
by Olesen et al. (2005) (dashed line in Fig. 5), in that there was an increase in the
degradation rate with the increase in fecal pellet surface:volume ratio, although the
degradation rates that we measured, exceeded the predicted rates in most cases,
particularly those produced with low Si diatom prey (Fig. 5). The generally higher
rates in our study might be caused by the higher temperature that we used when
compared with the previous studies (i.e., 23.5°C in our study *versus* 17°C and 18°C in
Olesen et al., 2005 and Hansen et al., 1996, respectively), but the role of cellular Si
content cannot be ignored.
The sinking rate of fecal pellets is usually considered to be related to their size
and density, which is in turn dependent on the concentration and composition of the
prey (Bienfang, 1980; Urban et al., 1993; Feinberg and Dam, 1998). We also
demonstrated that fecal pellet size, sinking rate and density are correlated with the
concentration of prey (Fig. 3B, 4), especially in the low silica diatom prey treatment.
Using the ratio of ingestion rate : fecal pellet production rate ratio as an index to
compare the diatom content per fecal pellet, no differences were found in pellets
produced from diets of the same silica content (Fig. 6), indicating that prey
concentration does not affect the package content of the fecal pellets. On the other
hand, copepods were shown to pack fewer hard-shelled (i.e., high Si) diatoms into
each fecal pellet in comparison to the soft-shelled (i.e., low Si) diatoms, although



these data were not significantly different statistically (Fig. 6).

The fecal pellets of copepods are formed in the midgut surrounded by a

peritrophic membrane, which is believed to protect the gut wall from the sharp edges
of the prey's cell wall. Moreover, the different sizes of fecal pellet with similar prey
content per fecal pellet is thought to result from the decreasing gut passage time with
the increasing of food concentration. A high prey concentration results in the food
passing through the gut more quickly and results in incomplete digestion, whereas a
low prey concentration allows the food to be kept in the intestinal tract for a longer
time and therefore digestion is more complete. We showed that the silica content of
the diatom cell wall determines the density and sinking rate of the fecal pellets when
the prey concentration was low due to complete digestion. In addition, we showed
that only the low concentration of low Si prey group, resulted in a significantly lower
fecal pellet density and sinking rate. In previous studies, the sinking rate and density
of the fecal pellets of *Calanus* were shown to be 70-171 m day$^{-1}$ and 1.07-1.17 g cm$^{-3}$,
respectively (Bienfang, 1980; Urban et al., 1993), which are considerably higher than
our results (Fig. 4). We suggest that these differences might be caused by the
differences in methodology used (Griffin, 2000).

To compare the combined effects of sinking and degradation rates for each

treatment, the reciprocal length scale, or L-ratio, which is the fraction of pellet
degradation per unit length traveled, was calculated (Feinberg and Dam, 1998). The
product of the L-ratio multiplied by the depth of the mixed layer can then be used to
provide the degree of degradation of a pellet within this layer. The results from such
calculation suggest that some diets might result in pellets that are substantially
recycled within the epipelagic layer whereas others result in pellets that are exported
out of the mixed layer in a relatively non-degraded manner. It should be pointed out,
however, the degradation rates we calculated are likely to be highly underestimated



due to the absence of zooplankton activities. For example, it has been reported that
copepod ingestion of entire fecal pellets (i.e., coprophagia) or the only partial break
down of fecal pellets might dramatically reduce the overall downward transport of
fecal material and thus increase its retention in the epipelagic layer (Lampitt et al.,
1990; Gonzalez and Smetacek, 1994; Svensen et al., 2012). For the same reason, plus
the absence of turbulence in our experimental set-up, our sinking rate measurements
are likely to be overestimated. Nevertheless, the L-ratio provides a relative indicator
of the export efficiency of the fecal pellets produced on diatom diets of different silica
content and can be used for a comparison with copepod fecal pellets produced with
other diets. Our results also show that pellets produced from high silica content
diatoms are more likely to sink out of the mixed layer before being degraded, when
compared with pellets from low silica content diatoms. On the other hand, fecal
pellets produced from a low concentration of prey with low Si content are the most
likely to be degraded in the euphotic layer (Table 3). Our results suggest that the
grazing activity of copepods might result in organic matter being mostly recycled in
the mixing layer during the fast growth period of diatoms (e.g., at the beginning of the
bloom), whereas it could accelerate the export of POC to the deep ocean by producing
fast sinking fecal pellets during the slow growth period of diatoms (e.g., during the
senescent stage of the diatom bloom).

In conclusion, the silica content of the cell wall of diatoms can affect the grazing

activity of copepods and influence the production rate, decomposition rate and sinking
rate of their fecal pellets. Our findings suggest that it is not only the nutritional quality,
but also the digestion process of copepods that can result in the different
characteristics of the pellets produced. In addition, it is a combination of both
degradation and sinking rates, (which are affected by the abundance and cellular silica
content of the diatom prey among other physicochemical factors), that determine the

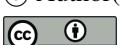


efficiency of the downward export of biogenic silica and organic carbon by fecal
pellets.


**Acknowledgements**
Financial support for this study was from the Research Grant Council of Hong
Kong (661809, 661610 and 661911) and the TUYF Charitable Trust (TUYF10SC08).

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





Table 1. Summary of the concentration and cellular silica content of the diatom prey
in each experiment.

| Expt. | Measurements | [Prey] | Silica level | Initial prey density (cells mL$^{-1}$) | Cellular silica (pg SiO$_2$ cell$^{-1}$) |
|---|---|---|---|---|---|
| 1 | | High | High | $8194 \pm 166.9$ | $55.7 \pm 1.7$ |
| | Fecal pellet | High | Low | $7976 \pm 8.5$ | $38.2 \pm 1.4$ |
| 2 | production | Low | High | $1640 \pm 28.3$ | $51.7 \pm 1.9$ |
| | | Low | Low | $1490 \pm 84.9$ | $31.4 \pm 6.6$ |
| 3 | | High | High | $8194 \pm 166.9$ | $55.7 \pm 1.7$ |
| | | High | Low | $7976 \pm 8.5$ | $38.2 \pm 1.4$ |
| 4 | Fecal pellet | High | High | $7499 \pm 63.6$ | $58.9 \pm 2.4$ |
| | degradation* | High | Low | $7344 \pm 169.7$ | $33.4 \pm 4.3$ |
| 5 | | Low | High | $1640 \pm 28.3$ | $51.7 \pm 1.9$ |
| | | Low | Low | $1490 \pm 84.9$ | $31.4 \pm 6.6$ |
| 6 | | High | High | $8114 \pm 138.0$ | $56.5 \pm 5.9$ |
| | Fecal pellet | High | Low | $7904 \pm 124.7$ | $27.0 \pm 0.6$ |
| 7 | sinking | Low | High | $1790 \pm 48.1$ | $52.1 \pm 1.3$ |
| | | Low | Low | $1545 \pm 75.0$ | $30.3 \pm 3.1$ |

The incubation time of the 3 fecal pellet degradation experiments can be found in
Table 3.




Table 2. Degradation rate of the fecal pellets produced by *C. sinicus* after they were
fed on diatoms with different silica content.

| Prey concentration | Incubation period | Silicon status of prey | n | Degradation rate (day$^{-1}$) |
|---|---|---|---|---|
| High | 48 hr | HSi | 3 | 0.21±0.15 |
|  |  | LSi | 3 | 0.91±0.17 |
| High | 24 hr | HSi | 4 | 0.03±0.04 |
|  |  | LSi | 4 | 0.15±0.02 |
| Low | 24 hr | HSi | 3 | 0.08±0.04 |
|  |  | LSi | 2 | 0.38±0.03 |

HSi: high silica content, LSi: low silica content.




Table 3. The L-ratio (m$^{-1}$), determined as the mean degradation rate constant (t$^{-1}$),
divided by the mean sinking rate (m d$^{-1}$), for each treatment.

| Prey silica content | High food concentration | Low food concentration |
|---|---|---|
| High Si | 3.91×10$^{-4}$ | 7.56×10$^{-4}$ |
| Low Si | 1.09×10$^{-3}$ | 1.65×10$^{-2}$ |












Fig. 1. The cellular silica content of *T. weissflogii* grown under different light
intensities. The error bars show one standard deviation (n=3).

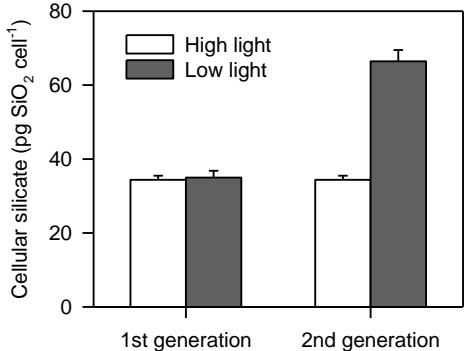






Fig. 2. Grazing rate (A) and clearance rate (B) of *C. sinicus* fed on diatoms with
different silica content. HSi and LSi are high and low silica diatom prey, respectively.
The error bars show one standard deviation (n=5).

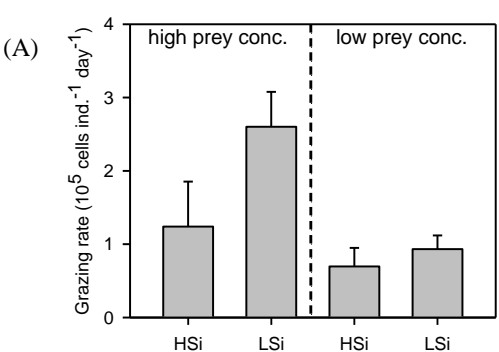

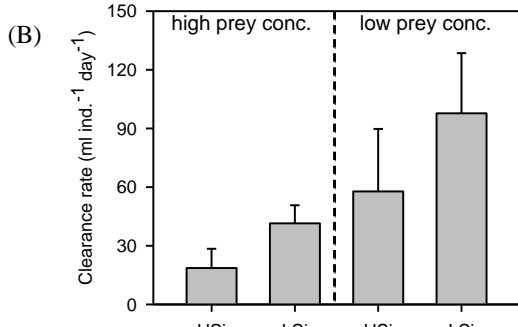






Fig. 3. The rate of fecal pellet production (A), and the average volume of each fecal

pellet (B), produced by *C. sinicus*. HSi and LSi indicate high and low silica diatom

prey, respectively. The error bars show one standard deviation (n = 5).

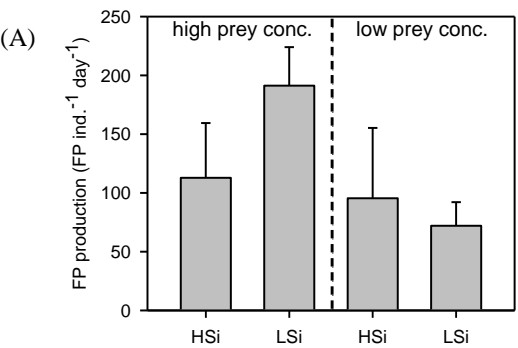

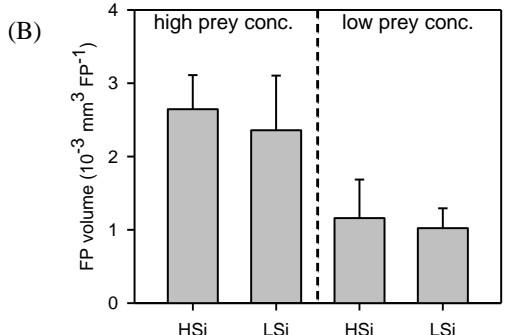







Fig. 4. The sinking rate (bars) and calculated density (open dots) of the fecal pellets
generated by *C. sinicus* produced following each treatment. HSi and LSi are high and
silica diatom prey, respectively. The errors bar show one standard deviation (n=3).

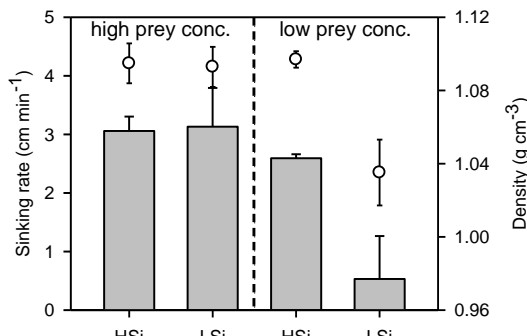




Fig. 5. The relationship between degradation rates and surface:volume ratio of fecal
pellets from different experimental treatments. HSi and LSi are high and low silica
content diatoms, respectively; high and low prey are high and low prey concentrations,
respectively; 48 hr and 24 hr are the incubation periods used for the degradation
experiments. The error bars show ±1 standard deviation and the dashed line shows the
relationship curve generalized by Olesen et al. (2005).

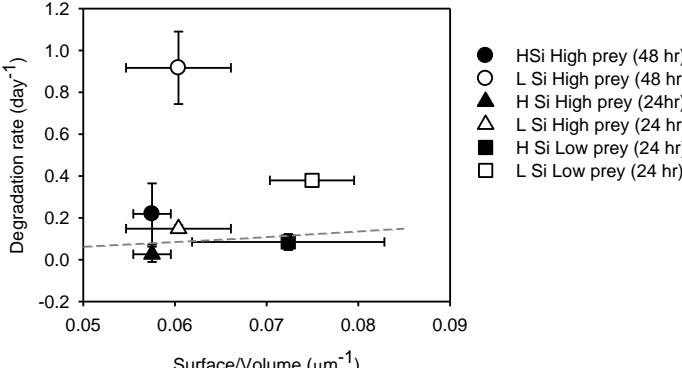





Fig. 6. The grazing rate: fecal pellet production rate ratio of each treatment. HSi and
LSi are the high and low silica diatom prey, respectively. The error bars show one
standard deviation.

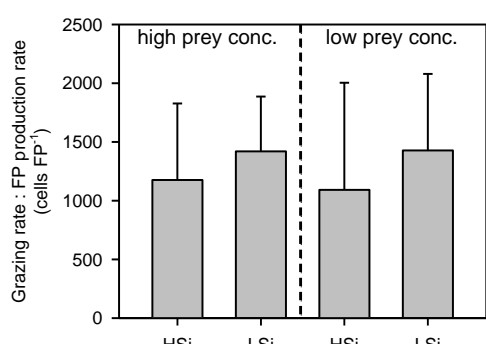
