# Peer review of "Effect of the silica content of diatom prey on the production, decomposition and sinking of fecal pellets of the copepod *Calanus sinicus"

_Biogeosciences, 2016_

## Referee Comment (RC1) · Anonymous Referee #1 · 25 May 2016

This is an interesting manuscript that should become acceptable after editorial revision. The manuscript addresses how silica content in diatoms fed to copepods can affect the production, decomposition and sinking rates of fecal pellets. In order to accept the conclusions of this manuscript, it must be demonstrated that silica contents of different diatom diets were actually different, and that these had been measured. While I do not doubt that this was the case, there is not enough information provided in the current version of the manuscript to allow readers to evaluate these critical aspects of the paper. For instance, on lines 89-91 it is stated that "Liu et al. (under review) recently demonstrated that the diatom Thalassiosira weissflogii, when grown at different

light levels, contains varying amounts of silica...". Considering that Liu et al. (under review) is unavailable to readers, and that no other information is provided on this critical aspect of the current manuscript, I suggest that the authors provide a bit more explanation of how these experiments were done, so that the results of the current manuscript become more understandable. Similarly, on lines 113-116, it is stated that "...the amount of biogenic silica in the diatom cells was measured using a modified version of the method described by Paasche (1980), following the procedures described more recently by Grasshoff et al. (1999)." Readers of the current manuscript should not have to stop reading here, and go find and read the two cited papers to understand the current manuscript. A brief bit of further explanation is required. Otherwise, most of the editorial corrections are in terms of consistency in the hyphenation of double-word adjectives, or more-than-three-word adjectives, and other minor grammatical and word-choice corections. These will be itemized below by line number.

19, 213, 221, 224, 236: hyphenate "high-silica"

20, 22, 120, 129, 152, 156: hyphenate "fecal-pellet"

21, 23, 139, 204, 212, 220, 235, 246, 321: hyphenate "low-silica"

22, 222, 238, 244, 247, 254, 335: hyphenate "low-prey"

22, 112, 139, 203, 209, 226, 242, 244, 246: hyphenate "high-"

29: hyphenate "highly-"

36: change "the main vehicle" to "a main vehicle"

39: change "Diatoms are the most abundant" to "Diatoms are among the most-abundant"

40: change "they represent the main component" to "they represent a main component"

52: hyphenate "pellet-production"

68: hyphenate "slowly-sinking"

91: change "contains" to "contain"

101: change "They" to "Copepods"

112, 213, 223, 229, 242, 251: hyphenate "low-"

175: hyphenate "loss-of-fecal-pellet"

200: hyphenate "first-"

200: hyphenate "second-generation"

203: hyphenate "high-light"

203: hyphenate "low-light"

209: hyphenate "low-cellular-silica"

213, 219, 239, 251, 333: hyphenate "high-prey"

220: hyphenate "significantly-higher"

227, 229: hyphenate "concentration-of-prey"

236: change "prey irrespective" to "prey, irrespective"

241: hyphenate "high-prey-"

248: hyphenate "high-silica-content"

249: hyphenate "low-silica-content"

254: hyphenate "low-silica'prey"

259: change "with the varying food concentration, followed" to "with varying food concentrations, and "followed"

259-260: provide clarification as to what was "a similar trend to that described in the

literature"

262: change "is one of the" to "has been suggested to be one of the"

266: hyphenate "weakly-silicified"

267: hyphenate "least-stable"

268: hyphenate "most-complex"

273: change "in this new study" to "in the current study"

276: change "in unit time and the ingestion rate" to "per unit time at a given ingestion rate"

278: change "and the ref. therein" to "and references therein"

279, 286: change "the fecal pellets" to "fecal pellets"

282: change "the pellets" to "fecal pellets"

283: change "they" to "fecal pellets"

284: change "significant differences" to "significant size differences"

289: change "they do not affect" to "these ratios did not affect"

293: change "and ref therein" to "and references therein"

295: change "mentions" to "have addressed"

295: change "rate" to "rates"

299: change "They showed that the fecal" to "Fecal"

308: change "they" to "our results"

310: change "the increase" to "an increase"

312: change "those produced" to "for fecal pellets produced"

312, 327, 339, 366: hyphenate "low-Si"

312: hyphenate "generally-higher"

320: change "are correlated" to "were correlated"

326: hyphenate "high-Si"

331: change "fecal pellet" to "fecal pellets"

332: change "is thought" to "are thought"

350: change "calculation" to "calculations"

353: change "the degradation" to "that the degradation"

355: change "coprophagia" to "coprophagy"

355: change "the only partial break" to "only partial break-"

363: hyphenate "high-silica-content"

365: hyphenate "low-silica-content"

369: change "mixing" to "mixed"

369: hyphenate "fast-growth"

371: hyphenate "fast-sinking"

371: hyphenate "slow-growth"

374: change "the production rate, decomposition rate" to "the rates of production, de-composition"

375: delete "rate"

379: change "determine" to "determines"

---

## Author Comment (AC1) · 5 Jun 2016

Referee #1 provided very detailed editing to our manuscript. Thank you very much!

Regarding to the two main issues, Liu et al. has been officially published (see reference below). We have also added more detailed procedures of measuring biogenic silica in the methodology section.

The reviewer suggested to put a hyphen between several pairs of words. While some of them are well accepted, I feel some words are better not being hyphenated. For example, "low prey concentration" is not exactly the same to "low-prey concentration".

[Figure]

Therefore, I have accepted some suggestion, e.g., high-silica and low-silica, but not all of them. Other than that, other English-related suggestions are all accepted or addressed.

Reference: Liu, H., Chen, M., Zhu, F., and Harrison, P.J.: Effect of diatom silica content on copepod grazing, growth and reproduction. Front. Mar. Sci., 3, 89, 2016. doi: 10.3389/fmars.2016.00089

---

## Referee Comment (RC2) · Anonymous Referee #2 · 15 Jun 2016

The present paper addresses effects of silica content of diatom prey on production rete and physical properties of fecal pellets of the copepod, Calanus sinicus. Since transport efficiency of materials transported by fecal pellet is a key mechanism of biological pump, BGD readers will be interested in its controlling mechanism. The results were novel and simple. And discussion is well organized. However, there are some specific points, which should be clarify before publishing.

Major points: A. Information of food quality of the two type of prey is insufficient. Contents of carbon and nitrogen should be presented in order to evaluate whether food

quality of the two type of prey other than silica content is similar or not. If authors has concluded that observed difference caused by the difference in cellular content of silica, food quality of two prey should be presented as much as possible (Probably contents of carbon nitrogen can be shown because C/N ratios of them are discussed). If available, cellular size of two preys should be presented (Probably possible, because authors used coulter counter to counts the number of prey).

B. It seems to be tricky to calculate of degradation rate using the number of intact and fragmented pellets. Authors have added half of the number of fragmented pellet to that of intact pellets. This assumption indicates that fragmented pellets at any degradation stage losses half of materials in the pellet. Does this assumption result in overestimation of the degradation rate in the early stage of degradation process? The overestimation will become remarkable, if material is hardly decomposable or if a rate obtained within short-time is extended to long-term change. Absolute value of the L-ratio in Table 3 must be carefully discussed, although qualitative relationships of degradability among four type of pellets will not change. Thus I doubt that most of the fecal pellets released in low prey concentration with low Si content will be degraded with in the euphotic layer (Lines 365-367). Additionally the higher degradation rate in this study than those in Hansen et al., 1996 and Olsen et al., 2005 may be caused by the counting procedure in this study. Because typical Q10 value of metabolic rate is around 2 (Kirchman and Rich, 1997: Microb Ecol 33:11–20 etc.), the difference in degradation rate is slightly high to explain by the difference in temperature.

Minor points: 1: Line 170: Product name and manufacturer of CCD should be presented.

2: Results: t-test has been used to compare the difference between two groups. But the difference among four groups was frequently discussed in discussion section. ANOVA should be used for the comparison. And the results of AOVA should be presented in figures 1-4 and 6.

3: Lines 336-338: Complete digestion? In the present method, authors cannot confirm whether prey is completely digested or not. The clearance rates suggests digestion under low prey concentration is more intensive than under high prey concentration.

---

## Author Comment (AC2) · 25 Jun 2016

We thank Referee #2 for her/his mainly positive comments to this manuscript.

I agree with the reviewer that other cellular properties of the prey may be equally important in order to test our hypothesis that the cellular Si content of diatoms affect zooplankton grazing, fecal pellet production, degradation and sinking. That is why we did not use two different species containing different amount of Si (to avoid difference in size and shape of the cells), or the same species growing at different growth phases (to avoid difference in cellular carbon and nitrogen contents). We did measure other

cellular properties for prey used in this study, but not for all experiments, because we found it is rather consistent if we raise the prey in the same condition. Since detailed information of prey growth condition and prey cellular characteristics has been reported in Liu et al. (2016), a paper uses the same method to obtain the same prey, but focuses on the impact of diatom prey on copepod reproductive physiology, we did not include those data here. To clarify that, the following sentence is added to this manuscript: "Other cellular parameters, such as cell size and carbon and nitrogen contents, were also measured for selected samples (data not shown), and the results were consistent with those reported in a previous study (Liu et al., 2016), which showed no significant difference between the two types of prey."

The second major point is related to the methodology for measuring the fecal pellet degradation rate. There is no doubt that the method used in this study, like all other available methods, is imperfect. The issue of under- or over-estimation during different stage of the degradation process that is raised by the reviewer is valid and we are aware of it. However, based on our experience, the pellets do not usually break into two half directly, but they are usually broken at one part and then lose its volume gradually. Since we do not count any debris smaller than half a pellet, we should have avoided double count, hence avoided severe overestimation.

The comments on the L-ratio is also well taken. I agree that the absolute value could be inaccurate, and it is affected by many biotic and abiotic factors, but the relative trend or pattern should be valid. Because the L-ratio can only serve as an indicator, we do not mean that "most of the fecal pellets released in low prey concentration with low Si content will be degraded with in the euphotic layer", but they are "the most likely to be degraded in the eutrophic layer" (should be mixed layer to be more accurate). We have already used a cautious tone in our discussion, and we have added more discussions accordingly to point out the potential issues related to the interpretation of the L-ratio.

Due to the same reason, temperature cannot explain the entire difference (in agreement with the reviewer) in the degradation rate estimated by this study and those from

previous studies. The experimental temperature in Hansen et al. (1996) and Oleson et al. (2005) were similar, but the latter reported higher degradation rates. We believe that the quantity and quality of the prey, particularly the cellular Si content in our case, plays a very important role in influencing the fecal pellet degradation rate. As to the issue of Q10, I believe it may not be as simple to just adopt the value of bacteria growth or production to that of fecal pellet degradation, because 1) bacteria can also use DOM, and 2) fecal pellets contain a lot of non-degradable (refractory) organic matter.

Minor points: 1) Product name and manufacturer of the CCD camera are added. 2) The reviewer is correct. However, although there are four treatments, we only compare the difference between two in most cases. Therefore, I have kept the use of t-test in the majority of the comparison, and used two-way ANOVA as suggested by the reviewer in some cases where the difference among four groups are compared, such as those reported in Fig. 3B and Table 2. 3) The reviewer is correct. This is just a speculation. We do not mean "complete digestion", but that the prey will be relatively better digested when the prey concentration is low. We added a word "relatively" to make it more accurate.

---

## Author Response (AR1)

[revised manuscript text omitted]